# Basket to Purkinje Cell Inhibitory Ephaptic Coupling Is Abolished in Episodic Ataxia Type 1

**DOI:** 10.3390/cells12101382

**Published:** 2023-05-13

**Authors:** Henry G. S. Martin, Dimitri M. Kullmann

**Affiliations:** Department of Clinical and Experimental Epilepsy, UCL Queen Square Institute of Neurology, University College London, London WC1N 3BG, UK; henry.martin@ucl.ac.uk

**Keywords:** episodic ataxia type 1, K_V_1.1 channel, ephaptic signalling, cerebellar cortex

## Abstract

Dominantly inherited missense mutations of the *KCNA1* gene, which encodes the K_V_1.1 potassium channel subunit, cause Episodic Ataxia type 1 (EA1). Although the cerebellar incoordination is thought to arise from abnormal Purkinje cell output, the underlying functional deficit remains unclear. Here we examine synaptic and non-synaptic inhibition of Purkinje cells by cerebellar basket cells in an adult mouse model of EA1. The synaptic function of basket cell terminals was unaffected, despite their intense enrichment for K_V_1.1-containing channels. In turn, the phase response curve quantifying the influence of basket cell input on Purkine cell output was maintained. However, ultra-fast non-synaptic ephaptic coupling, which occurs in the cerebellar ‘pinceau’ formation surrounding the axon initial segment of Purkinje cells, was profoundly reduced in EA1 mice in comparison with their wild type littermates. The altered temporal profile of basket cell inhibition of Purkinje cells underlines the importance of Kv1.1 channels for this form of signalling, and may contribute to the clinical phenotype of EA1.

## 1. Introduction

Episodic Ataxia type 1 (EA1) is a human channelopathy characterized by episodes of cerebellar incoordination. It is caused by heterozygous mutations of *KCNA1*, which encodes the Shaker-type potassium channel subunit K_V_1.1 [1]. In between attacks, patients frequently experience neuromyotonia, which is thought to arise from spontaneous action potentials in peripheral motor axons, consistent with juxtaparanodal clustering of K_V_1.1-containing channels [2]. The mechanisms underlying the paroxysmal cerebellar ataxia are however incompletely understood. K_V_1.1 is widely expressed in the CNS. The axon terminals of cerebellar basket cells in particular exhibit intense K_V_1.1 staining [2,3,4]. Consistent with the immunolocalization of K_V_1.1, the specific blocker Dendrotoxin-K evoked a broadening of action potentials recorded from synaptic varicosities of cerebellar basket cells in juvenile mice, although somatic action potentials were unaffected [5,6]. A potentially useful tool to link these findings to EA1 is a heterozygous mouse harbouring the EA1-associated mutation V408A [7]. Although this mutation is associated with only subtle loss of function effects in heterologous expression [8], action potentials recorded from basket cell synaptic boutons of the *Kcna1*^V408A/+^ mouse were significantly broader than in their wild type littermates [5].

In addition to forming boutons on Purkinje cell somata, basket cells extend protrusions that give rise to a specialized structure surrounding their axon initial segments, the so-called ‘pinceau’ (paint-brush). In this plexus, multiple basket cell axonal processes devoid of vesicles or sodium channels cluster together and form septate junctions [9]. An early proposal that this specialized structure compartmentalizes the extracellular space surrounding the Purkinje cell initial segment [10] has received support from the finding that action potential firing in basket cells of adult mice exerts a rapid inhibitory effect on Purkinje cell firing that resists blockade of GABA_A_ receptors [11,12,13]. This ephaptic mode of inhibition, when evoked by optogenetic stimulation of basket cells, is lost in mice lacking the K_V_1.1-interacting protein ADAM11 [14]. Given the enrichment of K_V_1.1 in the cerebellar pinceau, we set out to characterize both synaptic and ephaptic transmission, and asked how they are altered in the *Kcna1*^V408A/+^ model of EA1.

We find that, in wild type mice, ephaptic transmission exerts an early and short-lasting effect on spontaneous Purkinje cell activity under conditions where basket cells also fire spontaneously. In *Kcna1*^V408A/+^ mice, synaptic inhibition is unaffected but ephaptic inhibition is lost. The pinceau formation is larger in mutant mice than in their wild type littermates. We use live cell imaging to show that fluorescence corresponding to the pinceau predicts whether ephaptic inhibition is present at a given pair. Finally, we use optogenetic stimulation of basket cells to confirm that a brief ephaptic inhibition persists after blocking GABA_A_ receptors in wild type but not mutant mice. The findings reveal a striking dissociation between ephaptic and synaptic inhibition, confirm a critical role of K_V_1.1 in ephaptic signalling, and show that this ultra-rapid form of feed-forward inhibition is exquisitely sensitive to an EA1-associated mutation.

## 2. Materials and Methods

### 2.1. Cerebellar Preparation

EA1 mice were bred on a C57Bl/6J background by pairing heterozygous *Kcna1*^V408A/+^ males with stock females. Resulting pups were approximately 50:50 for the mutant allele. Mice were then group housed and maintained on a 12 h light-dark cycle with *ad libitum* access to food and water. All procedures and husbandry were performed in accordance with the UK Animals (Scientific Procedures) Act 1986 and approved by the Institute of Neurology Animal Welfare and Ethical Review Body (PPL PAF2788F5). 

Heterozygous *c-kit*^CreERT2^ mice [15] backcrossed onto a C57Bl/6J background were crossed with *Kcna1*^V408A/+^ to create an inducible molecular layer interneuron cell line. Mice were then crossed with homozygous reporter lines B6.Cg-*Gt(ROSA)26Sor*^tm14(CAG-tdTomato)^ (Ai14) or B6.Cg-*Gt(ROSA)26Sor*^tm32(CAG-COP4*H134R/EYFP)^ (Ai32). To reduce off-target expression [16], Cre was induced only once the mice were at least 5 weeks old with intraperitoneal Tamoxifen injection (100 mg/kg; sesame oil vehicle, Sigma-Aldrich (Gillingham, UK)). The resulting labelling was sparse, with <5% of molecular layer interneurons labelled.

Acute parasagittal cerebellar cortex slices were prepared from the vermis of *Kcna1*^V408A/+^ and control *Kcna1*^+/+^ littermates of both sexes between 7 and 12 weeks in age. To maintain spontaneous activity, a published protocol using N-methyl-D-glucamine (NMDG) was adapted [17]. Briefly, mice were anaesthetized with a Ketamine/Medetomidine mix, before perfusion with chilled NMDG(+) solution (in mMol; 35 NaCl, 75 NMDG, 2.5 KCl, 0.5 CaCl_2_, 7 MgSO_4_, 1.25 NaH_2_PO_4_, 26 NaHCO_3_, 5 HEPES.Cl, 25 D-Glucose, 5 Sodium Ascorbate, 2 Thiourea, 5 Ethyl pyruvate, 12 N-Acetylcysteine, 3 myo-Inositol, 2 Kynurenic acid; pH 7.5) oxygenated with carbogen. Parasagittal slices (260 μm) were prepared in chilled NMDG(+) solution from the isolated cerebellum with a Leica VT1200 vibrating microtome and allowed to recover for 12 min at 35 °C. Slices were then stored in a humidified interface chamber in oxygenated HEPES-based artificial CSF (110 NaCl, 2.5 KCl, 0.5 CaCl_2_, 3 MgCl_2_, 1.25 NaH_2_PO_4_, 26 NaHCO_3_, 10 HEPES.Cl, 25 D-Glucose, 5 Sodium Ascorbate, 2 Thiourea, 3 Ethyl pyruvate, 2 N-Acetylcysteine, 3 myo-Inositol; pH 7.4) for at least 45 min before recording.

### 2.2. Cerebellar Cortex Recordings

Fresh slices were anchored on an upright microscope stage (Olympus, Stansted, UK) and superfused (6–8 mL/min) with oxygenated artificial CSF (125 NaCl, 2.5 KCl, 2 CaCl_2_, 1 MgCl_2_, 1.25 NaH_2_PO_4_, 26 NaHCO_3_, 11 D-Glucose) at 33–34 °C. Both glutamatergic and GABAergic neurotransmission were left unblocked. Borosilicate patch pipettes were forged to give a tip resistance of between 3 and 5 MΩ and filled with 160 mMol NaCl. Basket cells were identified by infrared differential interference contrast (DIC) microscopy, as neurons lying within the third of the molecular layer closest to the Purkinje cell layer. Local Purkinje cells were then sampled in the same sagittal plane within a 500 µm interval drawn along the Purkinje cell layer centred on the location of the basket cell. Cells were maintained in the loose patch configuration (20 MΩ ≤ R_S_ ≤ 200 MΩ) in voltage clamp mode with the holding current set to 0 pA (usually between −5 and +2 mV). Purkinje cells usually fired tonically between 20 and 60 Hz. Basket cell activity was sparser and more irregular. Where basket cells fired at <1 Hz, they were transitioned to a giga-seal configuration and a depolarizing current was injected to increase their excitability [18]. Depending on slice preparation, up to 40% of basket cells failed to fire action potentials.

Membrane currents were low-pass filtered at 10 kHz and digitized at 50 kHz using a PCI-6221 (National Instruments, Austin, USA) board over a period of 10 min using a dual channel amplifier (Multiclamp 700B, Molecular Devices, Wokingham, UK). Data were recorded using custom scripts in LabVIEW (National Instruments), blind to *Kcna1* genotype. In a subset of experiments picrotoxin (100 µMol, Sigma) was added, allowing 3 min for equilibration. 

Recordings from molecular layer interneuron reporter lines followed a similar scheme with small modifications. In experiments with tdTomato or ChR2-EYFP labelling of basket cells, the axon was traced to the Purkinje cell layer to determine whether it extended to the pinceau. Before data acquisition a series of epifluorescence images were captured to allow post hoc scoring of pinceau presence. ChR2 was excited by an LED (470 nm) through the objective using a stimulation intensity found to evoke basket cell action potentials (1 ms, 1.25 mW/mm^2^). 

### 2.3. Pinceau Imaging and Immunofluorescence

Immunofluorescence experiments were performed blind to genotype. 3.7% paraformaldehyde (Sigma) perfused and fixed (24 h, 4 °C) cerebellar slices were mounted on a Leica VT1200 vibrating microtome and the vermis was sectioned into 50 μm slices. Slices were blocked (5% bovine albumin, 1% Goat serum, 0.3% Triton X-100 in Phosphate buffered saline) for 2 h, before overnight incubation in primary antibody (Table 1) in blocking buffer omitting goat serum. After washing, fluorescent labelled goat secondary antibodies were incubated with the sections before the final mount in Vectashield (Vector laboratories, Newark, NJ, USA). Confocal images were captured using a Zeiss LSM710 system. To allow relative quantification of pinceau K_V_1.1 immunofluorescence, a z-stack from a standard section of lobule V (anterior portion, 500 μm from dorsal surface) was captured (×40 objective); confocal and laser parameters were unaltered.

Confocal image stacks were analysed in Image J 1.53 (NIH). For quantification, all identified pinceaux in the standard section were analysed. PSD95 immunofluorescence was used to identify the z section with the maximal area of the pinceau and the mean K_V_1.1 signal was calculated from within this region. Somatic connections of sparsely tdTomato-labelled molecular layer interneurons were assayed by tracing axons from cells lying in the lower third of the molecular layer (putative basket cells). Presynaptic boutons synapsing onto calbindin-labelled Purkinje cell somata were visible as small tdTomato labelled varicosities. Pinceau-positive connections were scored where the axon then further extended into the granule cell layer boundary directly adjacent to the Purkinje cell axon initial segment. Image acquisition and quantification were performed blind to genotype.

### 2.4. Statistical Analysis

Paired basket and Purkinje cell activity was correlated by measuring the interval between the downward deflections of all Purkinje cell action potentials and each basket cell action potentials. For ChR2 recordings the onset of the light pulse was used as a surrogate for the Basket cell action potential. Intervals >100 ms or <−100 ms were considered as independent and discounted. Histograms were created from the interval arrays to show aggregate correlated activity. Where it was not possible to populate the average bin with >10 intervals the recording was not included. Pairs were scored as connected when the Purkinje cell activity decreased by >3 sd of the baseline firing in the 2–3 ms interval following the basket cell action potential. Correlated firing that did not show a reduction in Purkinje cell firing in this interval was scored as ‘other’. Histograms were constructed either to show absolute Purkinje cell firing rates or normalized by the baseline firing rate. Phase response curves were constructed from connected pairs using a correction for inter- spike interval sampling bias [19]. Data are given as mean ± standard error of the mean, graphed data show the 95% confidence intervals for individual points. Differences in control and *Kcna1*^V408A/+^ samples were tested for significance using Student’s *t*-test with a rejection of the null hypothesis upon a threshold significance of *p* < 0.05.

## 3. Results

### 3.1. Impaired Ultra-Rapid Basket Cell Mediated Inhibition of Spontaneous Purkinje Cell Activity in EA1 Mutant Mice

Purkinje cells exhibit pacemaker activity in vitro, which is modulated by spontaneous basket cell firing [20]. In the *Kcna1*^V408A/+^ mouse model of EA1, a broadened presynaptic action potential leads to an increased inhibitory drive onto Purkinje cells [5,7]. Consistent with these earlier findings, the coefficient of variation of inter-event intervals (IEI) of spiking Purkinje cells was greater in adult *Kcna1*^V408A/+^ mice than in wild type littermates (data acquisition and analysis performed blind to genotype), with a matching trend towards a lower firing frequency (Appendix A). In contrast, spontaneous activity in basket cells did not differ significantly between genotypes (Appendix A). Since both basket and Purkinje cells are spontaneously active, extracellular recordings from pairs of cells permits functional interrogation of the effect of the EA1 mutation without disruption of the intracellular milieu. We recorded in cell-attached mode from a basket cell in the molecular layer, and simultaneously from a Purkinje cell sampled randomly within a region extending 250 μm in either direction along the Purkinje cell layer, Figure 1a. Indirect measurement of the somatic action potential shape from the cell-attached current deflections failed to detect any genotype differences in *Kcna1^V408A/+^* basket or Purkinje cells (Appendix A). Basket-Purkinje cell connectivity was assessed by constructing time-binned histograms of Purkinje cell firing centred on basket cell action potentials. A decrease in the occurrence of Purkinje cell action potentials greater than three standard deviations of baseline, shortly following the basket cell action potential, was taken to indicate an inhibitory connection. By this measure, approximately 40% of randomly recorded pairs were connected. This estimate did not differ between wild type and *Kcna1*^V408A/+^ mice (Figure 1b; Fisher exact test, *p* = 0.73).

Focusing on connected pairs, a single basket cell action potential typically resulted in a ~2 ms pause in Purkinje cell firing, apparent in cumulative event plots constructed from 800 cycles centred on the basket cell event (Figure 1c). When the Purkinje cell activity was assessed in 1 ms bins, the average temporal profile of Purkinje cell firing was very similar between *Kcna1*^V408A/+^ and wild type pairs (Figure 1d). The trough in Purkinje cell firing rate was also very similar (absolute change in Purkinje cell firing frequency: wild type = −27.5 ± 2.0 Hz, *n* = 33; *Kcna1*^V408A/+^ = −27.8 ± 2.5 Hz, *n* = 22, data normalized by baseline firing shown in inset). Basket cell action potentials typically occurred at random intervals after any given Purkinje cell action potential. This permitted construction of a phase response curve (PRC) relating the change in timing of the next Purkinje cell action potential to the phase at which the basket cell action potential occurred. Purkinje cell firing was delayed in response to basket cell firing. The PRC exhibited a negative slope, consistent with a leaky integrator model; however, PRCs were similar for wild type and *Kcna1^V408A/+^* pairs (Figure 1e).

In many cases it was possible to record well over 1000 spontaneous basket cell spikes, together with postsynaptic Purkinje cell firing. This permitted reanalysis of the instantaneous Purkinje cell firing rate plotted at 200 μs intervals. We restricted attention to 26 out of 34 wild type and 19 out of 21 *Kcna1*^V408A/+^ basket-Purkinje cell pairs where 10 or more Purkinje cell action potentials fell on average in each 200 μs interval prior to the basket cell action potential (Figure 1f). As expected, the overall strength of Purkinje cell inhibition was again similar in *Kcna1*^V408A/+^ and wild type pairs. However, closer examination of the profile of inhibition revealed a small delay in the onset of inhibition in *Kcna1*^V408A/+^ pairs. Basket cell feed-forward inhibition of Purkinje cells is mediated by a mixed synaptic and ultra-rapid ephaptic mechanism, the latter partly mediated by K_V_1.1 and K_V_1.2 channels [11]. Because ephaptic inhibition is maximal approximately 1 ms after the basket cell action potential [11,14], we focused on this time interval. The instantaneous firing rate of Purkinje cells recorded from *Kcna1*^V408A/+^ pairs was less affected by a basket cell spike at 1.0 ms compared to wild type pairs (+1.0 ms interval change in Purkinje cell firing rate: wild type = −8.8 ± 1.4 Hz; *Kcna1*^V408A/+^ = −3.7 ± 1.9 Hz; *t*(43) = 2.19, *p* = 0.034). The effect persisted when the instantaneous Purkinje cell firing was normalized by the baseline activity prior to the basket cell action potential (Figure 1g). 

### 3.2. Cerebellar Pinceaux Are Enlarged in EA1 Mutant Mice

Inhibitory ephaptic coupling occurs in a plexus surrounding the Purkinje cell axon initial segment, the pinceau formation [11]. Within the wild type cerebellar cortex, this plexus is highly enriched for K_V_1.1 subunits (Figure 2a). Deficits in K_V_1 potassium channel function in the pinceau have been reported to affect ephaptic transmission [14]. Using PSD95 as an independent marker for the pinceau, K_V_1.1 expression was compared between *Kcna1*^V408A/+^ and wild type mice using immunofluorescence (Figure 2b). Consistent with normal membrane trafficking of K_V_1.1-V408A mutant channels [21], K_V_1.1 was enriched to a similar extent within the pinceau in both groups (standardized immunofluorescence signal: wild type = 72.7 ± 5.4 (*n* = 6); *Kcna1*^V408A/+^ = 77.8 ± 4.0 (*n* = 6); *t*(10) = 0.76, *p* = 0.46, Figure 2c. The pinceau area was however significantly greater in *Kcna1*^V408A/+^ than in wild type mice (Figure 2d; *p* = 0.023, unpaired t-test using three wild type and three mutant mice as the units of replication).

Basket cell axons variably extend collaterals to Purkinje cells to form pinceau formations [22], such that for any basket cell only a fraction of contacted Purkinje cells exhibit mixed ephaptic-synaptic coupling [11]. We asked if the degree to which basket and Purkinje cells are connected by pinceaux differs between the genotypes. An inducible Cre driver line at the *c-kit* locus has recently been show to allow selective reporter expression in cerebellar molecular layer interneurons without confounding Purkinje cell expression [16]. To assess the morphology and connectivity of basket cell axons, we used sparse expression of a tdTomato fluorescent reporter in *c-kit*^CreERT2^ mice (Figure 3a). Axons from individual basket cells were traced and scored for the presence of synaptic connections, apparent as varicosities surrounding Purkinje cell somata, and for pinceaux, identified as arborizations projecting through the Purkinje cell layer surrounding Purkinje axon initial segments (pinceau-positive) or terminating on the Purkinje soma (pinceau-negative, Figure 3b,c). (We verified that pinceau-forming axons invariably also synapsed onto the associated Purkinje cell body.) Basket cells innervated a similar number of Purkinje cells in wild type and *Kcna1*^V408A/+^ mice (Purkinje cells connected: wild type = 8.0 ± 1.1, *n* = 7; *Kcna1*^V408A/+^ = 6.6 ± 1.0, *n* = 7; *t*(12) = 0.91, *p* = 0.38). Furthermore, the proportion of soma-targeting axons that descended into the pinceau was also similar (proportion of pinceau-positive connections: wild type = 0.40 ± 0.05; *Kcna1*^V408A/+^ = 0.52 ± 0.08; *t*(12) = 1.18, *p =* 0.26). These data further argue that the EA1 mutation does not affect the innervation of Purkinje cells.

### 3.3. Pinceau Visualization Correlates with Ephaptic Coupling

The presence of both pinceau-positive and pinceau-negative connections implies that basket cells inhibit Purkinje cells either though mixed ephaptic + synaptic coupling or pure synaptic innervation. We asked if the strong, yet sparse expression of the tdTomato reporter in basket cell axons from the *c-kit*^CreERT2^ cerebellum allows visually guided paired recording, with concurrent classification of connections into pinceau-positive and pinceau-negative. 

We recorded spontaneous basket and Purkinje cell activity from pairs where a basket cell axon could be seen under epifluorescence microscopy to contact the Purkinje cell body (Figure 3d). In contrast to blind paired recordings (Figure 1), only one in 13 pairs failed to show a > 3 sd change in Purkinje cell firing in the 2–3 ms period following the basket cell action potential, implying that axonal tracing reliably identifies connected cells. We simultaneously scored connections as either pinceau-positive (*n* = 6) or pinceau-negative (*n* = 7). The maximal inhibition of Purkinje cell firing was comparable in pinceau-positive and pinceau-negative pairs (Figure 3e; Maximal change in Purkinje cell firing: pinceau-positive = −24.5 ± 7.0 Hz, *n* = 6; pinceau-negative = −20.7 ± 3.4 Hz, *n* = 7). This suggests that the presence of a pinceau does not in itself determine the strength of basket cell synaptic inhibition. In contrast, when the Purkinje cell firing was assessed 1 ms after the basket cell spike, a significant inhibition was only seen at pinceau-positive pairs (change in Purkinje cell firing at +1.0 ms: pinceau-positive = −7.8 ± 1.7 Hz, pinceau-negative = −2.4 ± 1.6 Hz; *t*(10) = 2.29, *p* = 0.043). This difference in ultra-fast Purkinje cell inhibition was equally apparent when normalizing by baseline Purkinje cell firing rates (Figure 3f). The instantaneous Purkinje cell firing rate at pinceau-negative connections was not significantly reduced from baseline 1.0 ms after the basket cell spike (*p* = 0.192).

These results, taken together with the deficit in ultra-fast inhibition at basket-Purkinje cell pairs recorded ‘blind’ in *Kcna1*^V408A/+^ mice (Figure 1f,g), argues that that the mutation selectively interferes with ephaptic coupling at the pinceau.

### 3.4. Evoked Basket Cell Firing Reveals Loss of Picrotoxin-Resistant Inhibition

A key feature of ephaptic inhibition is that it persists in the presence of GABA_A_ receptor antagonists. It proved difficult to maintain paired recordings for pinceau-positive connections for long enough to evaluate ultra-fast inhibition before and after pharmacological blockade of GABA_A_ receptors. We therefore turned to an optogenetic approach that has previously been used to characterize ephaptic coupling at the same connection [14]. We crossed a fluorescently labelled Channelrhodopsin-2-yellow fluorescent protein (ChR2-YFP) reporter line (Ai32) with the *c-kit*^CreERT2^ driver to permit specific expression of ChR2-YFP within a sparse population of basket cells. Brief (1 ms) optogenetic stimulation at threshold intensity evoked bursts of action potentials in ChR2-YFP expressing basket cells, and a decrease in spontaneous firing of Purkinje cells, which lasted longer than the pause seen with sparse spontaneous firing of basket cells (Figure 4a). This pattern is reminiscent of the effect of parallel fibre excitation of molecular layer interneurons in the intact cerebellum, which leads to a sustained period of Purkinje cell inhibition [23]. YFP-positive axons could again be traced to connected Purkinje cells, and scored for the presence of a pinceau. Neither the duration of inhibition nor its amplitude differed in relation to whether basket cell axons could be visualized extending to the pinceau (Figure 4b,c; duration of Purkinje cell firing rate decrease: pinceau-positive = 20.3 ± 1.7 ms, (*n* = 11); pinceau-negative = 18.8 ± 2.5 ms, (*n* = 7); trough in firing: pinceau-positive = −31.1 ± 2.3 Hz, pinceau-negative = −27.9 ± 3.5 Hz). 

The substantial temporal jitter of optogenetic basket cell activation (Figure 4a) prevents identification of ephaptic coupling on the basis of latency. Instead, we blocked GABA_A_ receptors with picrotoxin after first evaluating the synaptic component. Consistent with previous data on evoked basket cell firing [11], a picrotoxin-resistant ephaptic component of inhibition of Purkinje cell firing was only observed when basket cell axons descended into the pinceau (Figure 4d). Pinceau-positive Purkinje cells responded to optogenetic stimulation with an initial reduction in firing around 1 ms after the end of the light pulse, followed by a rebound as previously reported [11,14]. In pinceau-negative Purkinje cells, where connections were limited to the Purkinje cell soma, no inhibition was seen in the presence of picrotoxin (picrotoxin-resistant inhibition of Purkinje cell firing rate: pinceau-positive = −10.1 ± 3.1 Hz, pinceau-negative = 0.3 ± 1.4 Hz; *t*(15) = 2.71, *p* = 0.018). A comparison of synaptic and ephaptic signalling in the same pairs (estimated before and after addition of picrotoxin) showed a dissociation between the two components of inhibition (Figure 4e).

In parallel with the above optogenetic evaluation of ephaptic coupling, we expressed ChR2-YFP in a sparse population of basket cells in EA1 mutant mice, and restricted attention to Purkinje cells where YFP could be positively identified in the pinceau. Threshold optogenetic stimulation resulted in a sustained and strong inhibition of Purkinje cell firing in *Kcna1*^V408A/+^ mice and their wild type littermates (Figure 4f). The change in Purkinje cell frequency was similar in both groups (trough in Purkinje cell firing: wild type = −32.1 ± 1.6 Hz, *n* = 12; *Kcna1*^V408A/+^ = −32.9 ± 3.3 Hz, *n* = 9; baseline-normalized data in Figure 4g). Thus, similar to spontaneous basket-Purkinje cell paired recordings, the synaptic component of inhibition was unaltered in *Kcna1*^V408A/+^ mice. Ephaptic coupling was then isolated by blocking the synaptic component with picrotoxin. Purkinje cells in wild type mice, but not in *Kcna1*^V408A/+^ mice, exhibited a decrease in firing probability shortly after the optogenetic stimulus (Figure 4h, trough of Purkinje cell firing: wild type = −11.1 ± 1.7 Hz, *n* = 8; *Kcna1*^V408A/+^ = −2.9 ± 1.9 Hz, *n* = 7; *t*(13) = 3.15, *p* = 0.008; baseline-normalized data shown in Figure 4i). These data confirm a dissociation between ephaptic inhibition, which was absent, and synaptic inhibition, which was preserved, in the EA1 model.

## 4. Discussion

As the sole output from the cerebellar cortex, pathological Purkinje cell activity is implicated in a range of inherited or acquired forms of cerebellar ataxia. Nevertheless, there is a substantial heterogeneity in the mechanisms that result in Purkinje cell dysfunction, including channelopathies that directly affect their electrical [24] or dendritic properties [25,26], and mutations of other genes affecting synaptic transmission in the cerebellar circuitry [27]. Here we describe a new pathological locus affecting Purkinje cell function in EA1, the pinceau, the structural substrate of ephaptic transmission that critically depends on K_V_1 potassium channels [11,14]. 

K_V_1.1 is broadly expressed in the central and peripheral nervous system but is particularly concentrated at basket cell terminals and especially in their pinceau extensions. At basket cell synaptic boutons, the EA1-associated V408A channel mutation leads to a broadening of the presynaptic action potential, which is qualitatively similar to the effect of pharmacological inhibition of K_V_1.1-containing potassium channels [5]. Kv1.1 loss of function, in turn, is expected to increase calcium influx and GABA release, providing an explanation for an increase in spontaneous inhibitory postsynaptic currents and reduced paired-pulse ratio also reported in the same model [5,7]. The V408A mutation has also been shown to interfere with ‘analogue’ modulation of presynaptic action potentials by passively propagating subthreshold somatic depolarization in hippocampal neurons in culture [28].

Spontaneous Purkinje cell firing is shaped by background activity, including that of basket cells [20,29]. In the present study, in contrast to findings in juvenile mice, we did not observe a significant difference in Purkinje cell firing rate between genotypes, although the coefficient of variation of Purkinje cell inter-spike intervals was greater in *Kcna1^V408A/+^* than in wild type mice. Among possible explanations for this discrepancy are postnatal developmental and homeostatic changes in the properties of basket cells and their outputs [30]. Paired recordings of spontaneously active neurons permitted a direct assay of the effect of the heterozygous V408A mutation on basket cell—Purkinje cell synapse function. The probability of identifying connected pairs was, however, no different between wild type and *Kcna1^V408A/+^* mice, and, where connected, the inhibitory effect on Purkinje cell action potentials was of a similar strength. 

We observed negative-going phase response curves (PRCs), indicating a delay of the Purkinje cell spike following the basket cell action potential, which increased with the phase at which the basket cell action potential occurred. This pattern, consistent with a leaky integrator model of Purkinje cell firing, contrasts with positive PRCs obtained in response to somatic depolarizing current injection to Purkinje cells [19,31,32]. The shape of the PRC did not differ between wild type and *Kcna1^V408A/+^* mice. 

In contrast, the fine temporal resolution afforded by paired recordings revealed a striking difference in the onset of Purkinje cell inhibition in response to spontaneous basket action potentials. The decrease in Purkinje cell firing in the first millisecond following a basket cell action potential that was seen in the wild type cerebellum was profoundly attenuated in *Kcna1*^V408A//+^ mice. At this interval ephaptic coupling is expected to dominate over synaptic inhibition. 

The pinceau surrounds the axon initial segment of the Purkinje cell in close apposition, although a narrow sheath of glial processes is interposed at much of the interface [33]. In this environment a fast model of ephaptic inhibition has been put forward, whereby a passive depolarization propagates electrotonically from the basket cell axon into the pinceau, resulting in a biphasic capacitive current and an outward current mediated by a dendrotoxin-sensitive voltage-dependent potassium conductance. The resulting extracellular positivity surrounding the axon initial segment transiently increases the trans-membrane potential and inhibits firing in a narrow window approximately 1 ms after the basket cell action potential [11]. The loss-of-function effect of EA1-associated mutations of Kv1.1 is thus expected to interfere with ephaptic inhibition, as confirmed in the present study, and consistent with experimental manipulation of Kv1 channels by truncation of the interacting protein ADAM11 [14]. Immunofluorescence imaging of the pinceau in *Kcna1*^V408A/+^ mice also revealed a small but significant increase in its size. The pinceau has previously been reported to assemble normally without auxiliary K_V_β subunits or ADAM11 binding of K_V_1.1 [14,34]. The subtle structural difference detected in the present study is unlikely to contribute substantially to the loss of ephaptic inhibition.

The blind paired recordings from basket and Purkinje cells allowed the detailed temporal profile of inhibition to be studied without perturbing the intracellular milieu. However, extracellular recordings precluded post-hoc identification of the pinceau formation. Using the c-Kit promoter as a molecular layer interneuron marker, fluorescence labelling of basket cells axons suggested that approximately 50% of basket cell collaterals extended through the Purkinje cell layer, presumably innervating pinceaux. Thus, roughly half of the blind recordings can be expected to be from pinceau-negative connections, consistent with the relatively modest ephaptic coupling detected as an ultra-fast inhibition in the wild type cerebellum. However, this fast inhibition clearly depended on the presence of a pinceau, as revealed in the live imaging experiments, and confirmed with optogenetic stimulation of basket cells where a picrotoxin-resistant in inhibition was only seen in pinceau-positive connections. In these connections the ephaptic coupling was responsible for a brief reduction in Purkinje cell firing, which averaged 15%. This again is less than previously reported [14], but likely reflects the sparse expression of Channelrhodopsin in our experiments compared to others.

The magnitude of ephaptic inhibition did not correlate with the strength of synaptic inhibition. Why only some basket-Purkinje cell connections are equipped with ephaptic coupling is unclear, but the degree of pinceau elaboration has previously been reported to be maximal in zebrinII-negative regions and linked to Purkinje cell activity [35,36]. Importantly, recent transcriptomic data suggest that at least two types of molecular layer interneurons are found in the cerebellar cortex [37]. Both types can have basket cell morphology in the lower third of the molecular layer, but with distinct electrical properties and connectivity. The *c-kit* locus is active in both types of molecular layer interneuron; thus, our blind and reporter-guided recordings likely include data from both populations. However, pinceau formation appears to be unequal in these two types of molecular layer interneurons [37]. 

Neither the proportion of pinceau-positive connections, nor the total number of Purkinje cells contacted by a basket cell, was affected by the *Kcna1*^V408A^ mutation, arguing against the hypothesis that loss of a particular cell type is responsible for absence of ephaptic coupling in the mutant cerebellum.

Channelrhodopsin-driven basket cell excitation elicited a substantially larger and more prolonged period of Purkinje cell inhibition than when recording from spontaneously firing basket-Purkinje cell pairs, most likely because basket cells fire bursts of action potentials in response to a 1ms-long optogenetic depolarization [38]. Nevertheless, an ultra-fast component of inhibition persisted in the presence of picrotoxin only when basket cell axon collaterals were observed to extend through the Purkinje cell layer in the wild type cerebellum, further underlining the absolute dependence of non-synaptic inhibition on the pinceau formation. Conversely, no ultra-fast component of inhibition was observed in *Kcna1*^V408A/+^ Purkinje cells even with optogenetic stimulation of basket cells. These results closely resemble those reported in mice with an *Adam11* truncation that disrupts K_V_1.1/K_V_1.2 function [14]. However, in contrast to the *Adam11* truncation, enhanced presynaptic function at the *Kcna1*^V408A/+^ bouton is observed [5,7] which may account for the absence of a net increase in Purkinje cell output due to the loss of ephaptic inhibition.

Recent reports have expanded the range of connections engaging in ephaptic coupling in the cerebellar cortex, notably between Purkinje cells [39,40]. Interestingly, the ephaptic coupling is excitatory in the case of simple spikes, but inhibitory with complex spikes, the mechanism dependant on the type and location of ion channel engaged. However, these interactions appear to depend on capacitive and sodium currents but not on potassium currents, and so are unlikely to be affected in EA1.

The absence of pharmacological tools to interfere selectively with ephaptic signalling makes it difficult to directly link loss of this form of signalling with the clinical manifestations of episodic ataxia [41]. Current treatments for EA1 include the anti-seizure drug carbamazepine and the carbonic anhydrase inhibitor acetazolamide [42]. Whether these treatments directly affect ephaptic coupling remains to be determined. Among other research priorities is to determine how the loss ephaptic coupling alters the temporal profile of Purkinje cell output to the deep cerebellar nuclei [11,43] and information coding in the cerebellar cortex [44]. Similar to EA1, the *Adam11* mutant mouse, where K_V_1.1/1.2 is lost from the pinceau, responds to external stress with periods of ataxia [14]. However, the connection between ephaptic coupling and ataxia remains opaque. Ultra-fast feed-forward inhibition from basket cells is likely to support high temporal precision of Purkinje cell activity [43,45,46]. We speculate that some of this precision is lost in EA1. Impaired Purkinje cell pacemaker precision and response to parallel fibre stimulation is found in episodic ataxia type 2 (EA2) [47], linking these related disorders. However, the episodic nature of both EA1 and EA2 remains unclear.

## Figures and Tables

**Figure 1 cells-12-01382-f001:**
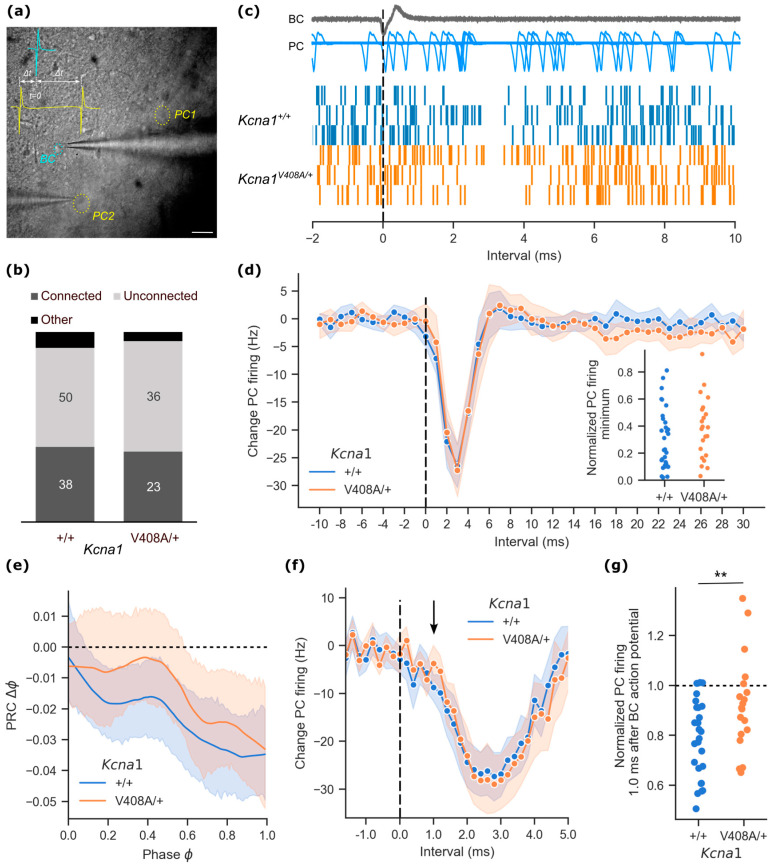
Ultra-fast basket cell inhibition of Purkinje cell activity is impaired in the *Kcna1*^V408A/+^ cerebellum. (**a**) Infrared differential interference contrast micrograph of cerebellar cortex slice with recording pipettes in cell-attached mode. Basket cell (BC) indicated by blue circle, Purkinje cells (PC1, PC2) by yellow circles (scale bar 40 μm). Inset. Scheme for recording relative timing of action potentials. All measurements are taken from the peak downward deflection of the soma-recorded event. (**b**) Relative proportions of connected and unconnected basket cell—Purkinje cell pairs in *Kcna1*^V408A/+^ and control cerebellum. Connected pairs were scored as recordings where the Purkinje cell exhibited a > 3 sd change in firing frequency in the period immediately following basket cell firing. Pairs with correlated firing outside of this period were scored as ‘Other’. (**c**) Representative cumulative event plots of Purkinje cell action potentials aggregated from 800 aligned basket cell events (dashed line) in three wild type and three *Kcna1*^V408A/+^ paired recordings. *Above.* Example overlaid traces (*n* = 100) from one experiment, aligned to the basket cell spike. (**d**) Mean change in Purkinje cell firing frequency from baseline plotted at 1 ms intervals relative to basket cell action potential (dashed line) from connected pairs (wild type *n* = 33, *Kcna1*^V408A/+^ *n* = 22; shading represents 95% CI). Inset: minimum in normalized Purkinje cell firing in the intervals immediately following basket cell firing in connected pairs. (**e**) Phase response curves (PRC), showing that Purkinje cells, on average, spiked later than expected following a basket cell action potential (wild type *n* = 24, *Kcna1^V408A/+^ n* = 17). (**f**) Data from pairs with high event counts, resampled to show Purkinje cell firing in 200 μs intervals relative to basket cell spike. Arrow highlights Purkinje action potentials occurring 1.0 ms after the basket cell spike. (**g**) Baseline-normalized Purkinje cell firing rate 1.0 ms after basket cell action potential from connected pairs (** *p* = 0.009).

**Figure 2 cells-12-01382-f002:**
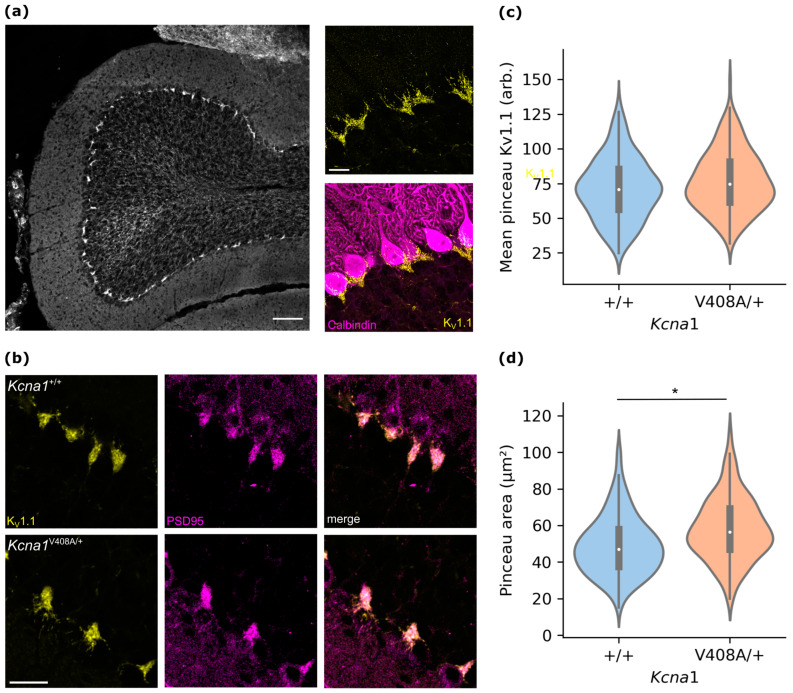
K_V_1.1 is concentrated in the pinceau in wild type and *Kcna1*^V408A/+^ cerebellum. (**a**) Low magnification image of cerebellar cortex folium section showing K_V_1.1 immunofluorescence (scale bar 100 μm). Right. High magnification confocal image showing concentrated K_V_1.1 immunofluorescence localized at the interface of Purkinje and granule cell layers. Right lower. Same image showing co-immunofluorescence labelling for the Purkinje cell marker calbindin (scale bar 20 μm). (**b**) Representative confocal sections of lamina adjacent to the Purkinje cell layer from *Kcna1*^V408A/+^ and control cerebellar cortex, labelled for K_V_1.1 (left, yellow) and PSD-95 (centre, magenta). Right, merged image (scale bar 20 μm). (**c**) Violin plot of mean K_V_1.1 immunofluorescence from PSD-95-outlined pinceaux (wild type *n* = 264, *Kcna1*^V408/+^ *n* = 213) from lobule V cerebellar vermal sections. Overlay. Box plot showing median and interquartile range, whiskers 1.5 * IQR from upper and lower quartiles. (**d**) Pinceau areas from PSD-95 immunofluorescence, * *p* = 0.023, wild type *n* = three animals, *Kcna1*^V408A/+^ *n* = three animals.

**Figure 3 cells-12-01382-f003:**
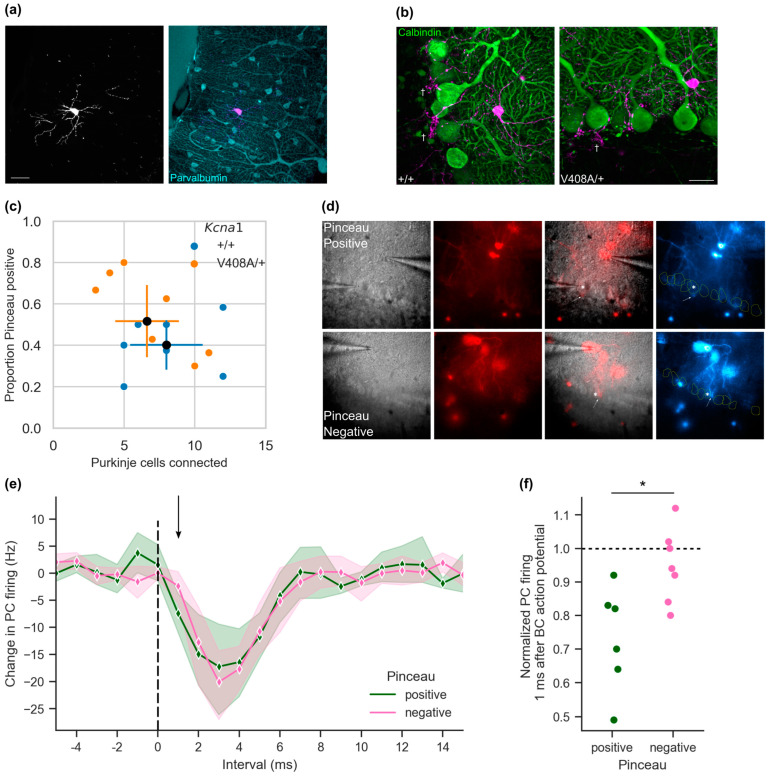
Ultra-fast inhibition correlates with the presence of a pinceau at basket—Purkinje cell pairs. (**a**) Sparse expression of TdTomato reporter with the molecular layer interneuron selective cKit-Cre^ERT2^ driver. Below. Immunofluorescence labelling of PV positive cells in cerebellar cortex molecular layer from the same section (scale bar 20 μm). (**b**) Calbindin immunofluorescence labelling of Purkinje cells in basket cell (magenta) axonal fields in *Kcna1*^V408A/+^ and wild type cerebellum. Examples of pinceau-positive descending axonal collaterals are indicated (†). (**c**) Scoring axonal contacts on Purkinje cells from sparsely labelled basket cells. (**d**) Example images from paired recordings showing pinceau-positive and pinceau-negative contact between basket cells (red) and selected Purkinje cells (*). Arrow indicates maximal projection of traced axon through Purkinje cell layer adjacent to selected Purkinje cell. Right. Pseudo-coloured image of basket cell fluorescence, Purkinje cell outlines in yellow from DIC micrograph (left panel). (**e**) Aggregated 1 ms histogram of Purkinje cell firing centred on basket cell action potential (t = 0) from pinceau-positive and pinceau-negative recorded pairs (pinceau-positive *n* = 6, pinceau-negative *n* = 7; shading represents 95% CI). (**f**) Normalized Purkinje cell firing frequency 1.0 ms after basket cell action potential (* *p* = 0.014).

**Figure 4 cells-12-01382-f004:**
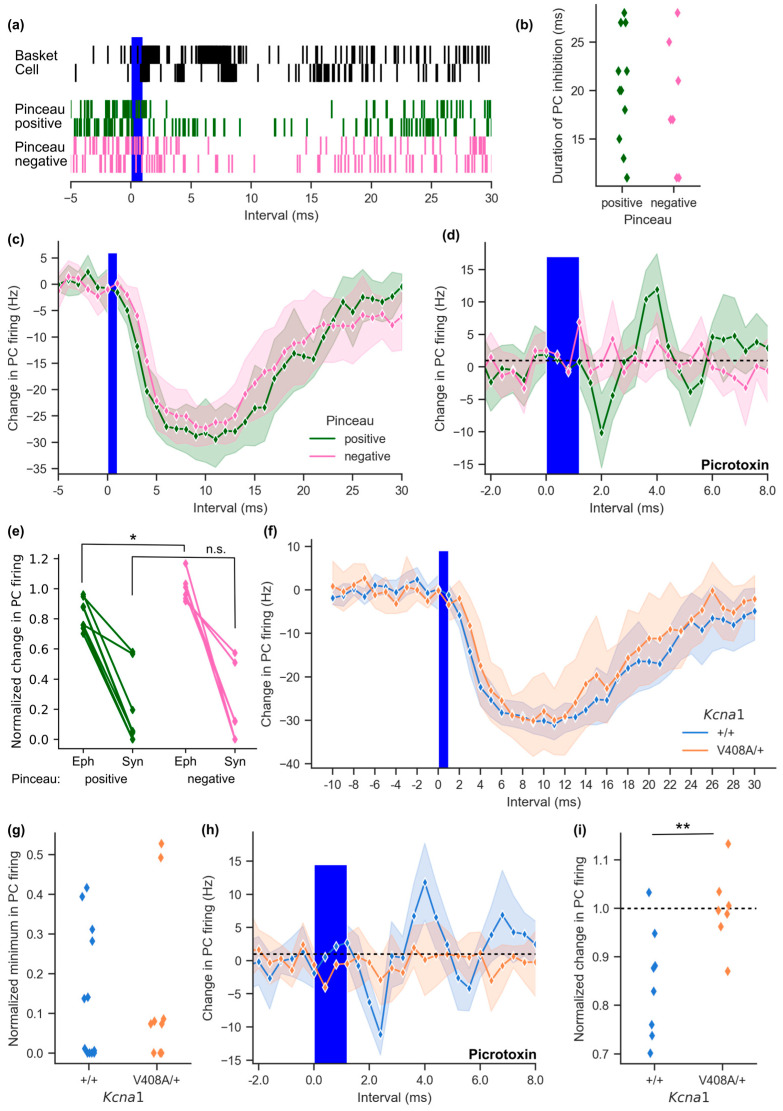
Basket cell ephaptic coupling is absent in the *Kcna1*^V408A/+^ cerebellum. (**a**) Cumulative event plots, constructed from 100 trials, showing ChR2-expressing basket cell activity (top) in response to threshold light stimulation. Below: connected Purkinje cell activity after pinceau-positive or pinceau-negative basket cell stimulation (2 example cells for each condition, 100 repeated trials). (**b**) Duration of Purkinje cell inhibition in response to basket cell light activation for pinceau-positive and negative pairs (interval with > 20% change in instantaneous Purkinje cell activity). (**c**) Plot of average change in Purkinje cell frequency following basket cell light stimulation in visually scored pinceau connections (pinceau-positive *n* = 11, pinceau-negative *n* = 7; shading represents 95% CI). (**d**) Instantaneous change in Purkinje cell firing frequency in 400 μs bins following basket cell light activation with the GABA_A_ receptor blocker picrotoxin (100 μM; pinceau-positive *n* = 8, pinceau-negative *n* = 6). (**e**) Baseline-normalized Purkinje cell firing 2 ms after basket cell light activation in the presence of picrotoxin (Eph), plotted with the normalized Purkinje cell firing rate prior to picrotoxin addition 5 ms after light activation (Syn). Lines connect data from individual Purkinje cells (* *p* = 0.018, n.s. = not significant). (**f**) Plot of light-induced change in Purkinje cell firing frequency in response to basket cell excitation in 1 ms intervals relative to light onset in *Kcna1*^V408A/+^ cerebellum (wild type *n* = 12, *Kcna1*^V408A/+^ *n* = 9). (**g**) Normalized minimum in Purkinje cell firing rate following basket cell optogenetic excitation in *Kcna1*^V408A/+^ and control cerebellum. (**h**) Purkinje cell activity at pinceau-positive connections from *Kcna1*^V408A/+^ cerebellum in the presence of 100 μM picrotoxin in response to optogenetic basket cell stimulation. Mean change in frequency plotted in 400 μs intervals (wild type *n* = 8, *Kcna1*^V408A/+^ *n* = 7). (**i**) Plot of baseline-normalized Purkinje cell firing rate for individual cells during the 2 ms interval following light-on at pinceau-positive connections (** *p =* 0.010).

**Table 1 cells-12-01382-t001:** Antibody details.

Epitope	Source	Number	Species	IF dilution
K_V_1.1	NeuroMab	K36/15	Monoclonal	1:1000
K_V_1.1	Alomone labs	APC-161	Rabbit	1:500
Calbindin	Swant	D-28k	Monoclonal	1:1000
Parvalbumin	Sigma	P3088	Monoclonal	1:1000
PSD-95	Abcam	AB2723	Monoclonal	1:1000

## Data Availability

Original datasets are available through UCL Research Data Repository: https://doi.org/10.5522/04/21747269 (accessed on 10 May 2023).

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
