# Peer review of "Basket to Purkinje Cell Inhibitory Ephaptic Coupling Is Abolished in Episodic Ataxia Type 1"

_cells, 2023, doi:10.3390/cells12101382_

Round 1

Reviewer 1 Report

In this paper Martin and Kullmann add knowledge to the mechanisms that contribute to altered cerebellar neurotransmission in episodic ataxia type 1. Using the Kv1.1V408A mouse, basket cells-Purkinje cells paired recordings, confocal microscopy and optogenetics they demonstrate reduced non-synaptic inhibition between BC-PC, occurring at the cerebellar pinceau, in mutant but not in WT mice. Thus, EA1 mutations, by affecting non-synaptic coupling at a region particularly rich in Kv1.1 channels, disrupt Purkinje cells activity and possibly contribute to the episodes of incoordination typical of EA1.

The paper is written very well; the results, obtained with a multidisciplinary sophisticated approach and many experiments performed, are robust and lead to the conclusion that the pinceau could be considered a new pathological locus affecting Purkinje cell function in EA1.

Here are some minor comments.

The authors show that basket cells inhibit Purkinje cells through mixed ephaptic and synaptic coupling. Through spontaneous and evoked paired BC-PC recordings they demonstrate unaltered synaptic inhibition but reduced pinceau-mediated inhibition of PC in Kv1.1V408A mouse. In addition, they describe no change in Purkinje cells spontaneous firing frequency, although an increase in Purkinje cell inter-event variability is reported. It appears that the V408A mutation affects mainly ultra-rapid ephaptic inhibition and this event appears to be the main dysfunction of cerebellar neurotransmission in Kv1.1V408A mouse. Can these findings merge with previous data showing increased duration of presynaptic potential at basket cells synaptic boutons and increased inhibitory post synaptic current at PC (Begum et al., 2016), to draw a EA1 pathogenetic hypothesis? This is not clear in the discussion.

The pinceau formation at AIS has been assumed to modulate PC outputs through electrical inhibition that is thought to be supported by enriched potassium channels and absence of sodium channels. EA1 patients benefit from sodium channels blockers, such as carbamazepine. Given the impaired BC-PC coupling described in the Kv1.1V408A mouse, the authors could comment on the mechanisms and therapeutic actions of these antiepileptic drugs.

Author Response

We are most grateful to the reviewer for a positive and encouraging review. Under the reviewer’s direction, we have made modifications to the manuscript to broaden discussion of the results and place our findings in a wider context. We hope the resultant manuscript will encourage future avenues of investigation of cerebellar dysfunction in EA1.

It appears that the V408A mutation affects mainly ultra-rapid ephaptic inhibition and this event appears to be the main dysfunction of cerebellar neurotransmission in Kv1.1V408A mouse. Can these findings merge with previous data showing increased duration of presynaptic potential at basket cells synaptic boutons and increased inhibitory post synaptic current at PC (Begum et al., 2016), to draw an EA1 pathogenetic hypothesis? This is not clear in the discussion.

We have expanded the Discussion to relate the findings of the present study to the earlier papers by Herson et al and Begum et al, which reported that GABAergic transmission at basket cell synapses was enhanced in the Kcna1V408A/+ model of EA1. Briefly, Begum et al reported a decrease in spontaneous Purkinje cell firing in Kcna1V408A/+ mice, which would fit with a net increased inhibitory drive. In the present study we only found a non-significant trend towards lower Purkinje cell spiking, when comparing Kcna1V408A/+ to wild type mice, although there was an increase in inter-spike interval variability. It is however difficult to relate these findings directly, because Begum et al studied juvenile mice, whereas the present study focused on adult mice, specifically because the pinceau formation continues to develop for several months after birth. Among other phenomena potentially contributing to the adult phenotype are homeostatic compensatory effects. In the discussion we now highlight the opposite effects of enhanced GABA release and impaired ephaptic inhibition, and include possible future directions to elucidate the downstream consequences of altered basket to Purkinje cell signalling.

The pinceau formation at AIS has been assumed to modulate PC outputs through electrical inhibition that is thought to be supported by enriched potassium channels and absence of sodium channels. EA1 patients benefit from sodium channels blockers, such as carbamazepine. Given the impaired BC-PC coupling described in the Kv1.1V408A mouse, the authors could comment on the mechanisms and therapeutic actions of these antiepileptic drugs.

We agree with the reviewer that a direct link between the effectiveness of sodium channel blockers in EA1 and the loss of ephaptic coupling endophenotype is not obvious. This could be extended to acetazolamide, which some patients find to be partially effective. We have modified the Discussion to introduce EA1 therapeutics and how, with the loss of ephaptic inhibition, Purkinje cells may be prone to increased excitation which could be a target of sodium channel targeting drugs.

Reviewer 2 Report

This is another elegant manuscript from the Kullmann group. His group conducts leading research in synaptic transmission. In this work, the authors discovered that the ultrafast inhibition caused by the pinceau between basket cells and Purkinje cell AIS is disrupted in an adult mouse model of EA1.

Major:

1, is there a possibility that Kv1 channels are also reduced in Purkinje cells in the mutant mice? The answer may be critical because the Kv1 current regulates Purkinje neuron spike shapes, firing frequency, and even the pause by the pinceau (https://doi.org/10.1523/JNEUROSCI.4358-07.2008). 

2, Purkinje neuron firing rates are lower and less regular in mutant rice than in WT mice. This seems contradicted by reduced ephaptic coupling in this work. If all background synapses are blocked in the slice, neurons should spike tonically and regularly (https://doi.org/10.1016/j.celrep.2018.07.011; https://doi.org/10.1016/S0896-6273(00)80379-7). The irregularity mainly comes from the background synaptic activation. 

3, Regarding Figure 1, the authors should analyze whether the firing rate reduction depends on the basket cell spike timing to avoid the potential bias brought by basket cell spike timing. The phase response curves of Purkinje neurons are firing rate dependent, and the phase of stimulus is critical for shifting the next spike timing. (https://doi.org/10.7554/eLife.60692;  https://doi.org/10.1371/journal.pcbi.1000768).

4, This work will be further improved if the authors make the experimental protocol more realistic. In the brain, basket cells are activated by parallel fibers that also excite Purkinje cells. A protocol considering the FFI loop will help to understand the importance of the pinceau in a more biologically relevant way (https://doi.org/10.1523/JNEUROSCI.1719-20.2020; https://doi.org/10.1113/jphysiol.2004.075028). At least, this should be discussed as a limitation in the discussion.

Minor

5, In Figure 1, it will help if the authors provide some spike traces for panel c. Panel c is confusing. The authors need to clarify what are the raster plots. Intuitively the Purkinje neurons fire at 1000 Hz, which should be different from what the authors tried to convey.

6, the changes of the ultra-fast inhibition between WT and mutant mice are not pronounced. How can deep cerebellar nuclei tell these tiny differences? The authors should discuss this.

7, Regarding Figure 3, the authors should briefly explain the meanings of pinceau-postive and pinceau-negative. They are defined in Methods but never mentioned in the main text.

8, can the authors explain the pause near 10-15 ms in Figure 4a?

Author Response

We are grateful to the reviewer for their very helpful comments. We have substantially modified the manuscript to address these critiques with new analysis of Purkinje cell phase response curves and major additions to the text.

Major comments

  1. Is there a possibility that Kv1 channels are also reduced in Purkinje cells in the mutant mice? The answer may be critical because the Kv1 current regulates Purkinje neuron spike shapes, firing frequency, and even the pause by the pinceau.

We agree with the reviewer that any change in the Purkinje cell action potential shape could potentially affect our measurement of the timing between basket and Purkinje cell action potentials in Kcna1V408A/+ mice and affect our paired recordings. They were however obtained in cell attached mode, and so we only have an indirect handle on spike shape. The waveforms recorded in cell attached voltage-clamp mode are, moreover, sensitive to the pipette location relative to the AIS and the quality of the seal, precluding direct integration of the signal. Nevertheless, we have looked at the spike waveforms and they were broadly comparable between experiments. From these intervals we can infer basic somatic Purkinje cell action potential kinetics. We find that these measures are broadly unchanged in Kcna1V408A/+ neurons. In Figure S2, we now provide quantification of the cell attached biphasic response in both the Purkinje and basket cell action potentials.

  1. Purkinje neuron firing rates are lower and less regular in mutant rice than in WT mice. This seems contradicted by reduced ephaptic coupling in this work. If all background synapses are blocked in the slice, neurons should spike tonically and regularly. The irregularity mainly comes from the background synaptic activation.

As pointed out by the reviewer the principal source of irregularity in Purkinje cell firing comes from synaptic molecular layer interneuron inhibition, particularly in the ex vivo state. The default experimental set-up in our study has been to record a mixed synaptic and ephaptic inhibition with both GABAergic and glutamatergic signalling left unblocked; the exception being Figure 4d&h where a pure ephaptic inhibitory signal was isolated. Prior work has shown that the V408A mutation enhances inhibitory synaptic transmission at basket to Purkinje cell synapses (https://doi.org/10.1038/ncomms12102), and significantly lower Purkinje cell firing rates were reported. However, that study was performed in slices from juvenile mice, while the present study used adult mice because the pinceau formation only develops slowly. Enhanced synaptic inhibition in the adult Kcna1V408A/+ cerebellum could account for the increased coefficient of variation in Purkinje cell interspike interval that we report. Unexpectedly, we did not directly detect this enhanced inhibitory synaptic transmission in our paired basket-Purkinje cell recordings. The optogenetic experiments are more difficult to interpret in this regard, because the synaptic inhibition saturates. Ultimately, loss of inhibitory ephaptic coupling in the Kcna1V408A/+ cerebellum may partially compensate for enhanced GABAergic inhibition, resulting in an attenuation of the expected change in Purkinje cell firing. We have modified the Discussion to expand on this, and added a sentence to the Methods stating that glutamatergic and GABAergic signalling was not blocked.

  1. The authors should analyze whether the firing rate reduction depends on the basket cell spike timing to avoid the potential bias brought by basket cell spike timing. The phase response curves of Purkinje neurons are firing rate dependent, and the phase of stimulus is critical for shifting the next spike timing

The reviewer raises an interesting point about Purkinje cell firing. To our knowledge Purkinje cell phase response curves have not previously been studied in response to basket cell input. Using our paired recordings data from ‘connected’ pairs of basket and Purkinje cells, we have plotted Purkinje cell phase response curves (PRCs) for wild type and Kcna1V408A/+ mutants in response to basket cell input (Figure 1e). Broadly the PRC is similar in both groups. Interestingly, it has a downward slope consistent with a leaky integrator model of Purkinje cell firing.

Basket cell action potentials were uniformly distributed (< 10 % deviation) across the Purkinje phase in all save 4 experiments. We tried excluding these experiments from the PRC analysis, however the resultant PRC were not materially different. Therefore, for simplicity we have included all recordings in our analysis.

We now include the PRC in the manuscript and the abstract, because it is arguably an important addition to the literature.

  1. This work will be further improved if the authors make the experimental protocol more realistic. In the brain, basket cells are activated by parallel fibers that also excite Purkinje cells. A protocol considering the FFI loop will help to understand the importance of the pinceau in a more biologically relevant way

We agree that the wider effects of changes in ephaptic signaling in the Kcna1V408A/+ cerebellum remain to explored. Our motivation in working on a reduced preparation was first to isolate the ephaptic effect, but also secondly to avoid possible confounds from parallel fiber stimulation in the measure of ephaptic signaling (https://doi.org/10.1101/001123). Nonetheless there remain many possible extensions to this study. We now include discussion of the limitations this study and highlight possible future directions.

Minor comments

  1. 5. In Figure 1, it will help if the authors provide some spike traces for panel c. Panel c is confusing. The authors need to clarify what are the raster plots. Intuitively the Purkinje neurons fire at 1000 Hz, which should be different from what the authors tried to convey.

We have added to panel 1c example spike traces leading the reader to how the plots are constructed. For clarity we now refer to these plots as ‘cumulative event plots’ rather than the misleading raster plot.

  1. The changes of the ultra-fast inhibition between WT and mutant mice are not pronounced. How can deep cerebellar nuclei tell these tiny differences? The authors should discuss this.

We agree that the ephaptic effect that we record is indeed small. However, the experimental set-up is arguably biased towards an underestimate of its amplitude, because the paired recordings of spontaneously active basket and Purkinje cells were blind to the presence of a pinceau structure, meaning pinceau negative connections are included in our analysis of Purkinje cell inhibition (Fig. 1f,g). Meanwhile our optogenetic experiments used sparsely transfected c-kit basket cells, which differs from previous experiments (https://doi.org/10.1523/JNEUROSCI.1346-15.2015) where a stronger ephaptic coupling was recorded. We now include discussion of the weaker ephaptic coupling found in this study compared to previous reports. Ultimately, the relevance of ephaptic inhibition to information transmission in the cerebellar cortex may rest on its temporal profile rather than its overall amplitude.

  1. Regarding Figure 3, the authors should briefly explain the meanings of pinceau-postive and pinceau-negative. They are defined in Methods but never mentioned in the main text.

The terms pinceau-positive and pinceau-negative have now been explained in the main results section.

  1. can the authors explain the pause near 10-15 ms in Figure 4a?

We think that the change in profile of Purkinje cell inhibition in the optogenetic experiments compared to spontaneous basket cell firing reflects the propensity of the basket cells to fire a burst of high frequency action potentials in response to light stimulation (the example basket cell cumulative event plots show cells firing preferentially two or three action potentials respectively followed by a brief refractory period). The effect is to extend the period of reduced Purkinje cell firing for up to 25 ms from the light pulse. In some experiments the inhibitory effect of the presumed second and third action potentials could be resolved as a change in Purkinje cell firing with each action potential. However, since the basket cell response was variable this effect is averaged out over many experiments. The effect did not differ between wildtype and Kcna1V408A/+ recordings, leading us to focus on the fast ephaptic inhibition.

Round 2

Reviewer 2 Report

The authors have solved all my concerns.